# Outcome of Communication Training in Veterinary Studies: Influence on the Perception of the Relevance of Veterinary Competencies and Self-Assessment of Communication Skills

**DOI:** 10.3390/ani13091516

**Published:** 2023-04-30

**Authors:** Mahtab Bahramsoltani, Sonja Bröer, Susann Langforth, Corinna Eule, Alina Prior, Lena Vogt, Ting-Ting Li, Rebecca Schirone, Alina Pohl, Katharina Charlotte Jensen

**Affiliations:** 1Institute of Veterinary Anatomy, School of Veterinary Medicine, Freie Universität Berlin, Koserstraße 20, 14195 Berlin, Germany; 2Institute of Pharmacology and Toxicology, School of Veterinary Medicine, Freie Universität Berlin, Koserstraße 20, 14195 Berlin, Germany; 3Institute of Food Safety and Food Hygiene, School of Veterinary Medicine, Freie Universität Berlin, Königsweg 67, 14163 Berlin, Germany; 4Centre for Veterinary Clinical Services, Veterinary Hospital, School of Veterinary Medicine, Freie Universität Berlin, Oertzenweg 19b, 14163 Berlin, Germany; 5Veterinary Skills Net, School of Veterinary Medicine, Freie Universität Berlin, Oertzenweg 19b, 14163 Berlin, Germany; 6Clinic of Animal Reproduction, School of Veterinary Medicine, Freie Universität Berlin, Königsweg 65, 14163 Berlin, Germany; 7Institute for Veterinary Epidemiology and Biostatistics, School of Veterinary Medicine, Freie Universität Berlin, Königsweg 67, 14163 Berlin, Germany

**Keywords:** veterinary studies, communication trainings, communication skills, veterinary competencies, self-assessment, e-learning, simulation training

## Abstract

**Simple Summary:**

Communication is of great importance in all veterinary fields. An e-learning course in the second year and a practical course based on role plays in the third year were set up in the veterinary studies program at Freie Universität Berlin. This study investigated the effects of the communication courses on the assessment of the skills that veterinarians need in their professional work and on the students’ self-perception of their own communication skills. Students conducted the survey before and after the e-learning course as well as before and after the practical course. Additionally, veterinarians were surveyed regarding which professional skills seem relevant from their point of view. The results show that students who have just completed the practical course rated communicative competencies for veterinary professional success higher than the other groups. These students also rated their own communication skills higher than all other students. This indicates that the practical course in combination with the e-learning course they had attended in their second year helped them to improve their self-assessment of communication skills, which is something they will benefit from in their professional lives.

**Abstract:**

Since communication skills contribute significantly to professional success among veterinarians, there is a particular focus on developing communication classes in veterinary curricula. At Freie Universität Berlin, an e-learning course covering the basics of communication and a practical communication course based on role plays with and without simulation persons have been established. The outcome of these communication courses on the assessment of the relevance of several veterinary competencies and on the self-assessment of communication skills using the SE-12 questionnaire was investigated. For this purpose, students were surveyed before and after the e-learning course as well as before and after the practical course. Veterinarians were also surveyed on the relevance of veterinary competencies. The relevance of communicative competencies for professional success was rated significantly higher by the students after completing the practical course than by the other students and the veterinarians. Self-assessment of communication skills showed little increase after the e-learning course, but a significant increase after the practical course. Thus, an effective outcome of the communication classes was observed mainly after the practical course. However, the effect of the e-learning course cannot be ruled out since the students participating in the practical course have also completed the e-learning course beforehand.

## 1. Introduction

The majority of veterinary students desire to work in general (46%) or specialist (12%) clinical practices [1]. Usually, students decide to study veterinary medicine because they want to work with animals, have had exposure to companion animals or livestock, have veterinary professionals as role models, and/or have a fondness or talent for the natural sciences [2]. Special competencies such as interpersonal and business skills do not typically play a role in the decision-making process to become a veterinarian or to attend pre-veterinary courses [3]. However, the students might realize during the course of their degree program that communication is one of the most important skills for a medical practitioner [4]. Adequate and successful communication in the medical setting significantly contributes to better compliance and adherence to treatment plans and, consequently, treatment success, a good doctor–patient relationship, as well as the healthcare professional’s well-being [5]. In Germany, the importance of communication skills in medical professions has been acknowledged by implementing communication training as a core subject into the Medical Studies Masterplan to allow students to be prepared for efficient patient-physician communication in their clinical practice [5,6]. Similarly, communication skills are listed by the European Association of Establishments for Veterinary Education (EAEVE) as “Day One Skills” for veterinarians [7]. These skills include effective communication with different audiences such as animal owners, employees, colleagues, business partners, and the public. In addition, veterinarians should be able to actively listen and communicate comprehensibly adapted relevant information to the target audience [8]. Indeed, communication skills are regarded as being among the most important key competencies for veterinarians and define subsequent success in veterinary practice [9]. However, interaction and communication with pet owners are a particular challenge for veterinarians [10,11], and a perceived lack of communication skills leads to the experience of cognitive dissonance and stress, which can lead to mental disorders [11]. Conversely, it has been shown that learning and training communication skills lead to a lower stress experience and are associated with higher professional success for veterinarians [9,10]. Nevertheless, education in communication is not yet a mandatory part of veterinary teaching in Germany; no competencies are anchored in the ordinance for the licensing of veterinarians (TAppV) [12]. The urgent need to include communication in TAppV and in the German veterinary curriculum has been demonstrated by Gruber et al. (2021) [13].

At the School of Veterinary Medicine at Freie Universität Berlin, different modules for teaching communication skills have been integrated into compulsory modules over the last five years: The theoretical foundations of communication, such as the 4-sides model [14], transactional analysis [15], strategies of conversation [16], and the Calgary-Cambridge observation guide [17], are taught in the second semester (first year) in the form of a mandatory e-learning course [18]. A newer addition is a practical communication course, in which communication and conversation skills are trained within small groups in various scenarios from the veterinary profession in role plays with and without trained simulation persons (actors or actresses). This training takes place for students in the fifth semester (third year).

The aim of this study was to determine the impact of these communication courses on the students’ awareness of the relevance of communicative competencies in the veterinary profession and their self-assessment of their own communication skills.

## 2. Participants, Materials and Methods

### 2.1. Participants

For this study, different groups of veterinary students at Freie Universität Berlin on the one hand and veterinarians in Germany on the other hand were surveyed. The online surveys were conducted between November and December 2022 via the online survey application LimeSurvey Version 3.28.21^®^. All individuals were informed of the purpose of the study, that their participation in the study was voluntary, and that their data would be collected anonymously. All participants gave their informed consent for inclusion before they participated in the study.

The first group consisted of students enrolled in the first semester (first year). These students represented the participants who had not yet participated in any communication course as part of their studies. The second group of students were third-semester students (second year). These students had taken the mandatory e-learning course on the basics of communication (see below for description) in the previous semester. The survey of these two groups of students took place during the dissection courses in the first and third semesters, respectively, in which all students in the semester participated. The students were informed about the content and objectives of the study at the beginning of the respective course and were then given the QR code to take part in the survey. The third group was represented by the students in the fifth semester (third year). These students were surveyed at the beginning and at the end of the mandatory practical communication course (see below for description). The last group were veterinarians. The questionnaire for this group was distributed to various groups on social media.

### 2.2. Communication Courses

#### 2.2.1. E-Learning Course on the Basics of Communication

The e-learning course, which has been mandatory for students of the second semester since 2019, was created with the authoring tool tet.folio Version 2019 [19] and made available via the learning management system Blackboard Learn Version 9.1 Q4 2019^®^ [20], which is used at Freie Universität Berlin.

The beginning of the e-learning course is designed to raise awareness of the topic. In several videos, students see bad practice examples, e.g., an exam situation, a veterinary consultation interview, as well as statements from pet owners about the quality of veterinary consultations. In addition, students have the task of searching for good and bad ratings of veterinarians online, in which pet owners address the communication skills of veterinarians.

The first communication model introduced is Schulz von Thun’s 4-sides model [14]. Students will use a variety of videos to identify potential communication faults at the factual level and learn the method of paraphrasing as part of active listening [21]. Next, the appeal level is addressed, which consists of the aspect of a message for which the sender tries to influence the receiver [14]. To understand the effects of appeals, students are asked to indicate the extent to which they would like to comply with appeals, which are expressed in different ways. Subsequently, the students have the opportunity to see how their fellow students responded. Following it, the students are then invited to indicate how an appeal should be phrased for them to like or dislike following it. The unit concludes with several videos of situations involving veterinary consultations, whereby students are asked to identify positive and negative aspects of the formulation of each appeal. To understand self-disclosure in communication situations, students learn about methods that people use on the one hand to impress (“impression techniques”), or on the other hand to hide unpleasant aspects of themselves (“facade techniques”) [14]. In this unit, students also practice recognizing the nature of self-disclosure using different video examples. The meaning of the relationship level is introduced with a self-reflection exercise. The students are first presented with the statement of a pet owner as a text and are asked to assess the pet owner’s age, gender, occupation, appearance, and socioeconomic status. Afterwards, they can view the corresponding assessments of their fellow students. In the next steps, they listen to the same statement first in an audio file and then in a video. When assessing the same parameters again, the students recognize possible changes in their assessments. This exercise serves to realize how much the assessment of people is determined by the projection of one’s own unpleasant behaviors and the transfer of previous experiences with other people [14]. The unit ends with lessons on the phenomena of “projection” and “transfer” [14] as well as on the meaning and value of empathy in communication. The learning objectives in this module are to recognize the impact of structure, conciseness, and comprehensible language on the factual level, to get to know convincing strategies on the appeal level, to perceive behaviors on the level of self-disclosure as “impression and facade techniques” and to manage them, as well as to become aware of the role of “projection” and “transfer” on the relationship level [14].

The second module of the e-learning course is about transactional analysis [15]. The first unit focuses on the different ego states, i.e., parent ego state, adult ego state, and child ego state [15], which are introduced in animation videos. Following each video, the students are presented with various statements from the field of veterinary study and profession that they are to assign to the respective ego state. At the end of the unit, the students have the task of responding to a statement by a veterinarian as a pet owner from different ego states. The second unit on transactional analysis deals with the different types of transactions (complementary, crossed, and ulterior [15]), which are also introduced in animation videos. In the associated exercises, students are asked to match statements in such a way that complementary or crossed transactions are created and then to determine from which ego state the respective statements originate. Additionally, they are to identify the ulterior messages in various statements as well as specify from which ego state these ulterior messages originate and to which ego state they are addressed. Moreover, several dialogues are presented where students are asked to indicate the type of transaction using respective arrows. The unit on transactional analysis ends with an application example from veterinary practice. The learning objectives in this module are to identify verbal and non-verbal elements in communication that indicate the different ego states, as well as to recognize complementary, crossed, and ulterior transactions and to apply strategies to avoid crossed and ulterior transactions [15].

The third module is focused on the methods of conversation. The first unit introduces the five stages of conversation [16]. Afterwards, the students have the task of assigning sentences to the individual stages of the conversation in different conversational situations, i.e., in a study group, in a team, or in a veterinary consultation. In the second unit, the students get to know the Calgary-Cambridge observation guide (CCOG) [17]. The subsequent exercises are focused on the phases of the veterinary consultation as defined in the CCOG, i.e., preparation, initiation, gathering information, explanation and planning, and closure [17]. For the phase “preparation”, students have to decide whether different aspects and activities belong to this phase or not. For the phase “initiation”, the exercise consists of students evaluating different ways in which the veterinarian welcomes pet owners presented in videos. Since asking questions is of particular significance in the phase “gathering information”, the tasks for this phase refer to the different types of questions and in which situation each type of question is appropriate. The phase “explanation and planning” is addressed in a task in which students assess whether different aspects and actions contribute to the success of this phase. In addition, the principles of participatory decision-making are introduced [22,23]. “Closure”, as the last phase, is presented with two different video representations of the end of a veterinary consultation. The students’ task is to identify positive aspects of a successful closing phase in the videos. The learning objectives in this module are to learn about an optimal structure for the veterinary consultation by the CCOG, recognize what belongs to a good preparation of the consultation, which behaviors contribute to an appreciative welcome of the pet owner, formulate open and closed questions appropriately, bring about participatory decision-making, as well as to shape the closure of the consultation to mutual satisfaction [17,22,23].

The e-learning course ends with a knowledge test in the form of a multiple-choice quiz that students complete as a self-test.

#### 2.2.2. Practical Communication Course

The practical communication course has been mandatory for fifth-semester students since 2022. The course is based on a flipped classroom concept [24] and includes e-learning units for preparation as well as four lessons, 90 min each. In each lesson, a case is worked on that deals with communicating in a situation of the veterinary profession. In two cases, the task consists of anamnesis (cat with conjunctivitis, mastitis case on a dairy farm), the third case is about breaking bad news (equine asthma), and the last case focuses on a conflict situation (improper use of an electric prod by an animal transporter at the abattoir). In the lessons on anamnesis, the learning objectives are to apply the CCOG to take a structured anamnesis, to create a trusting atmosphere for conversation, and to ask targeted questions to obtain relevant information [17]. The learning objectives of the lesson on breaking bad news are to explain the pet’s disease in accordance with the pet owner’s prior knowledge, to communicate the treatment options in a precise and understandable way, and to bring about participatory decision-making with the pet owner [25]. In the lesson on conflict situations, the learning objectives are to adequately assess the conflict situation [26], understand the perspective of the animal transporter and thus the cause of his misbehavior, and apply verbal and non-verbal strategies for de-escalation [27] in order to make the animal transporter realize that his behavior was not appropriate. The cases were developed in cooperation with the team “Special Teaching Formats” of the Charité-Universitätsmedizin Berlin, who also conducted the training of the simulation persons as well as a workshop to qualify the lecturers for the delivery of the lessons [28,29].

For all cases, the e-learning material for preparation includes information on medical and legal background knowledge. Additionally, students receive e-learning material for each case that addresses the respective communicative competencies. In preparation for the anamnesis, an animated video was used to refresh their knowledge of the CCOG [17]. For the equine asthma case, the six-step protocol for delivering bad news (SPIKES) [25] was introduced with a poster. To prepare for communicating in the conflict situation, students were provided with an animated video on Glasl’s escalation stages [26] and a poster on verbal and non-verbal de-escalation strategies [27], with a special focus on the change of perspective. Also included in the e-learning material for all cases are the rules for giving and receiving feedback [30].

The lessons (concept and schedule adapted from [28]) take place in groups of about 20 students each. Each lesson starts with the lecturer explaining the conversation setting and focus to the students, as well as the learning objectives. In the next step, the students’ prior knowledge acquired in the e-learning phase is activated by giving them cards with keywords related to the case, which they have to explain. Students who wish to perform the role play are then identified. In the two cases for the anamnesis as well as for the conflict situation, a simulation person takes on the role of the animal owner or animal transporter, so that only one student is selected for the role of the veterinarian. In the role play involving breaking bad news, all roles are taken on by students: the veterinarian, animal owner, and stable owner. The remaining students are observers during the following role play and are given specific observational tasks focusing on the following aspects: conversation structure, problem exploration, relationship building, empathy/understanding the other person’s perspective, and arranging the end of the conversation. Then the setting for the role play is prepared, with dummies serving as animals. In cases where a simulation person is involved, this person joins the lesson at the beginning of the role play. During the role play, the lecturer switches to the role of an observer. The role play can only be interrupted and stopped by the students participating in the role play with a previously agreed time-out sign, in order to receive tips if they do not know how to proceed. After the role play, the observers and the simulation person write down and structure their feedback. When everyone has finished, the student who played the veterinarian is given the opportunity to express his or her own perception of the conversation situation. Subsequently, he or she receives 360° feedback, starting with the simulation person or the fellow students involved in the role play, followed by the observing students, and finally by the lecturer. After the joint discussion of the case and possible clarifications of open questions, the lesson ends with the lecturer revisiting the learning objectives and formulating the corresponding take-home messages.

### 2.3. Questionnaires

At the beginning of the online survey, before taking the questionnaires, third-year students who were surveyed at the end of the practical communication course were asked to indicate whether they participated in the role plays as a veterinarian, animal owner, or solely as an observer. The veterinarians were asked to indicate the number of years they have worked as a veterinarian (<2 years, 2–5 years, 6–10 years, >10 years).

A self-developed questionnaire (questionnaire 1) was used to obtain an assessment of which competencies the respondents perceived as relevant to the veterinary profession. Within this questionnaire, participants were asked to select exactly five from a list of potential veterinary competencies (defined by the authors) that they consider most relevant to professional career success (Table 1). This questionnaire was used with both students and veterinarians.

Another questionnaire (questionnaire 2) was used to assess students’ self-efficacy regarding their own communication skills (Table 2). For this purpose, the SE-12 questionnaire, which was developed for assessing clinical communication competencies for healthcare professionals [31], was used. The questionnaire was translated into German and adapted to veterinary medicine by replacing “patient” with “patient owners” (a common term for owners of companion and farm animals in Germany) in all cases. The participants rated the statements on a 10-point Likert scale from 1 (very uncertain) to 10 (very certain); alternatively, the statement could also be rated as “not relevant” [31].

### 2.4. Statistical Analysis

Data was exported in Microsoft Excel 2019^®^. The data of the people who clicked on the link but did not answer any questions were deleted (n = 29). As the students in the third year were invited to participate before and after the practical communication course, we planned to merge the answers by an individual code of five letters, consisting of the third letter of the month of birth of their mother, the third letter of their month of birth, the second letter of their mother’s first name, the second letter of their first name, and the second letter of their birthplace. However, 91 answers could not be matched to a corresponding code. Moreover, a few codes did not explicitly match another code. Due to concerns about data loss, we did not match the answers and regarded them as independent in the following.

The analyses were conducted with IBM SPSS Statistics Version 27^®^. The assessment of the competencies (questionnaire 1) was displayed in cross tables stratified by group. For each competency, Pearson’s chi-square tests were calculated. For the self-assessment of communication skills (questionnaire 2), descriptive analyses were carried out, again stratified by group. As the scores were not always distributed normally, Kruskal–Wallis tests were calculated. For the pairwise comparison of the groups, *p*-values were adjusted for multiple testing (Bonferroni correction). The significance level was set at 5%. The results are presented as bar charts for questionnaire 1 and boxplots for questionnaire 2 using Microsoft Excel 2019^®^.

## 3. Results

### 3.1. Number of Participants

A total of 618 participants were included in the analysis of questionnaire 1. They were distributed as follows. From the first-year students, 144 students participated out of a total of 192 students, and from the second-year students, 101 students participated out of a total of 181 students. Out of the total of 149 third-year students, 106 completed the questionnaire before the practical communication course and 97 at the end of the course. Among the 97 third-year students surveyed at the end of the course, 24 students (25%) reported being involved in the role plays as veterinarians, 14 students (14%) confirmed being involved as animal owners, and the remaining 59 students (61%) reported being observers only. From the group of veterinarians, 160 people participated in this survey. Of these, 12 participants (8%) had <2 years of work experience, 38 individuals (24%) had 2–5 years, 32 individuals (20%) had 6–10 years, and 77 individuals (48%) had >10 years; one person did not specify his or her work experience.

Questionnaire 2 was used to survey students only and was completed by a total of 419 students. These were 121 first-year students and 93 second-year students. Of the third-year students, 108 students participated in the survey before the practical communication course and 97 at the end of the course.

### 3.2. Assessment of the Relevance of Competencies

Both veterinary students (first-year, second-year, and third-year before and after the practical communication course) and veterinarians were asked to select from a list of 15 competencies, which they considered to be the five most relevant for a successful career of veterinarians. Since there were no significant differences by role (veterinarian, pet owner, or observer) among the group of third-year students surveyed after the practical communication course, these students were included as one group in the analyses. Similarly, there was no significant relationship between the length of professional experience and the assessment of the relevance of the competencies among the veterinarians (except for “being tidy”). Therefore, the group of veterinarians was not divided according to professional experience in the further analysis.

Regarding the top five competencies, there were almost no differences between the surveyed groups. For all groups surveyed, the top five included the competencies of being good with people, being good with animals, asking targeted questions, and being a good listener. For the second-year and third-year students, the competencies of taking the perspective of the other person and resolving conflicts were also included. Among the top five were the competency of being up to date with the latest knowledge for the first-year students and the competency of structuring for the veterinarians. However, there were differences between the groups regarding individual competencies.

Figure 1 shows the percentage of people in each group who selected or did not select each competency for all 15 listed competencies. Table 3 displays the results of the chi-square test. For four competencies, there was no significant difference between the group of respondents and the assessment of the relevance of the competency. Two of these competencies, i.e., being good with people and the competency of asking targeted questions, were given high relevance by all groups. The competency of always having specialist knowledge at hand was considered relevant by around 20–36% in all groups. The least relevance was attributed to the competency of being good at mental arithmetic, as only nine respondents chose this competency. Due to the sparse data, no calculations were carried out for the competency of being good at mental arithmetic.

For the assessment of the other competencies, significant differences existed between the groups. The relevance of being good with animals was rated as relevant by 81–91% of all groups except the third-year students after the course (62%). The relevance of being a good listener was rated highest by third-year students before the course (64%), followed by second-year students (56%), third-year students after the course (54%), veterinarians (50%), and first-year students (50%). The relevance of arguing convincingly and the relevance of resolving conflicts were rated significantly higher by third-year students after the course (50% and 69%) than by the other groups (10–37% and 25–36%). Interestingly, the veterinarians rated these two competencies higher than the other groups of students. Moreover, the relevance of taking the perspective of the other person was rated highest by third-year students after the course (59%), followed by second-year students (44%). The relevance of structuring was rated highest by veterinarians (54%) and was less often chosen by students (19–36%). The relevance of being up to date with the latest knowledge was rated highly by second-year students (43%) and first-year students (41%), but less often by veterinarians (33%), third-year students before the course (29%), and third-year students after the course (18%). The relevance of having manual skills and the relevance of having technical skills were rated highest by first-year students (31% and 11%), followed by veterinarians (28% and 5%), but were less often chosen by the other groups of students. The relevance of using medical terminology declined within veterinary education: 16% of first-year students but only 1% of veterinarians chose this competency. The relevance of being tidy was chosen more often by first- and second-year students (12%) and fewer by the other groups (2–7%).

### 3.3. Self-Assessment of Communication Skills

For an overall analysis of students’ self-assessment of communication skills, a total score summed across items was calculated for each participant (Figure 2). Generally, the students assessed themselves quite favorably, with a median score of 7.3. However, the score differed significantly between the groups of students (chi-square (3) = 18.5, *p* < 0.001, n = 419). The pairwise comparison illustrated that the third-year students’ self-assessment of communication skills after the practical communication course was significantly higher than that of the first-year students (*p* = 0.004), the third-year students before the practical course (*p* = 0.001), and slightly higher compared to the second-year students (*p* = 0.055). The other comparisons showed no significant differences.

However, the results differed between the individual items of the questionnaire (Figure 3). Generally, the students assessed themselves worse concerning the planning of the conversation (median = 7, mean = 6.4) and structuring of the consultation (median = 7; mean = 6.5) and better concerning showing empathy (median = 8; mean = 7.9), active listening (median = 8, mean = 7.8), and encouraging the patient owners to express their thoughts and feelings (median = 8, mean = 7.8). For six out of twelve items, the Kruskal–Wallis test revealed differences between the groups concerning the self-assessment of the respective communication skill (Table 4).

The first skill was structuring the conversation with the patient owners. For this skill, the third-year students rated themselves significantly higher than the other groups. Also concerning the second skill regarding checking the patient owners’ understanding of the information given, the third-year students after the practical communication course rated their abilities higher than the other groups of students. Concerning the next three skills (identify the issues the patient owners wish to address during the conversation, make an agenda/plan for the conversation with the patient owners, and clarify what the patient owners know in order to communicate the right amount of information), the self-assessment was significantly higher among third-year students after the practical course than among first-year students and third-year students before the course. However, third-year students after the course did not rate themselves significantly higher than second-year students concerning these three skills. The sixth skill was to make a plan based on shared decisions with the patient owners. Self-assessment in this skill was only significantly higher among third-year students after the practical course compared to third-year students before the course, but not compared to the other two groups.

Regarding the other skills, i.e., urging the patient owners to expand on their problems/worries, listening attentively without interrupting or changing of focus, encouraging the patient owners to express thoughts and feelings, demonstrating appropriate nonverbal behavior, showing empathy and acknowledging the patient owners’ views and feelings, and closing the conversation by assuring, that the patient owners’ questions have been answered, the groups’ self-assessment of the respective communication skill did not show any significant differences.

## 4. Discussion

As evidenced by numerous studies, it is undisputed that communication skills are among the key competencies of veterinarians [4,7,32,33]. For this reason, all veterinary schools in Germany are working to establish and expand curricula for teaching communication skills, even though communication teaching is not yet required as per the TAppV [18]. Thus, an e-learning course to teach the basics of communication for first-year students and a practical communication course for third-year students were introduced as mandatory courses at the veterinary school of Freie Universität Berlin in 2019 and 2022, respectively. This study examined the possible effects of the skills taught in these communication courses on perceptions of the relevance of different competencies in the veterinary profession and on self-assessments of communication skills.

With the first questionnaire, students and veterinarians were asked to select from a list of 15 competencies the five they considered most relevant to a successful veterinary career. The list included competencies from the area of methodological and self-competencies that the authors, as veterinarians, considered potentially relevant. The question was whether the students who completed the communication courses (e-learning and practical course) gave a higher relevance to the competencies related to communication and to what extent the students’ assessment approached those of the veterinarians, assuming that veterinary professional activity sensitizes veterinarians to the relevance of communicative competencies.

Two competencies that were rated as very relevant by all groups were “being good with people” and “being able to ask targeted questions”. Studies that surveyed veterinarians also found that they perceived having high empathy and compassion for people as highly relevant competencies for veterinarians [34,35]. Moreover, this assessment increased with work experience [34]. Although the differences were not significant, the competency of being good with people was rated higher by second- and third-year students than by first-year students who had not yet had communication classes. Therefore, the experiences in the communication courses may have increased the awareness of this aspect. Observations of veterinary consultations show the high value of targeted questions, both open-ended and closed-ended, for obtaining information [36]. This fact was apparently recognized by the students surveyed separately from communication classes. The assessment of the competencies of “arguing convincingly”, “resolving conflicts”, and “taking the perspective of the other person” differed significantly in the surveyed groups. These competencies were assessed as more relevant by the third-year students after the practical communication course than by the other surveyed groups. Veterinarians are constantly faced with the challenge of convincing pet owners of the strategies for treating their animals and the associated costs with plausible arguments [37,38]. The fact that the relevance of “arguing convincingly” was rated higher by the third-year students after the practical communication course could be due to the focus of the role play dealing with equine asthma in the course. In this role play, the task was to convince the animal owner and the stable owner of the necessary changes in the husbandry of the horse. During a focus group survey of veterinarians who were new to the profession, the skill of resolving conflicts was defined as one of the most important skills [39]. The management of a conflict situation was the focus of the role play on the improper use of the electric prod at the abattoir, demonstrating to the students the necessity of this skill for professional activity. This could be the reason why the competency “resolving conflicts” was rated the highest by the students after the practical communication course. Especially in difficult consultation situations in veterinary practice, the competency to take the perspective of the other person and thus develop empathy and compassion contributes to results that are satisfactory for both sides [40]. This became particularly comprehensible to the students in the role play on the improper use of the electric prod at the abattoir and could thus have led to the students attributing a very high relevance to this competency after the practical communication course. For effective communication, it is also important that veterinarians are good listeners [35,41]. The students who had completed the e-learning course on the basics of communication but had not yet participated in the practical communication course attributed a higher relevance to the competency “being a good listener” than the students after the practical course. In the e-learning course, the technique of active listening [21] was introduced and deepened in related exercises. Even though listening was part of the communication in the practical course, this aspect was not directly addressed again. Therefore, it can be assumed that other newly introduced communication techniques were more prominent for third-year students after the practical course. The competency of structuring is of particular significance in veterinary consultation [17]. For this purpose, the CCOG [17] was already introduced to the students in the e-learning course, refreshed in the e-learning preparation material for the practical communication course, and practically applied in the two role plays on anamnesis. Amongst the students surveyed, the relevance of this competency was rated highest among third-year students after the practical communication course, followed by third-year students before the practical course. However, the relevance of structuring was rated lowest by second-year students, thus lower than by first-year students. Therefore, it can be suggested that it was only with repetition, and even more so through practical application of the CCOG [17], that the relevance of structuring became apparent. Since the relevance of the competency of structuring was rated highest by the surveyed veterinarians, there is a tendency for the students to approximate the assessment of the veterinarians during their studies.

The competency of “being good with animals” is not only considered extremely relevant by pet owners but is also rated very highly by veterinarians [35]. Similarly, among the competencies not related to communication, the relevance of “being good with animals” was rated highest overall in this study. However, the rating of relevance was significantly lower among third-year students after the practical communication course. Furthermore, after having completed the practical communication course, the third-year students were the only group surveyed who ranked being good with animals lower than being good with people. This could be related to the intensive work on communication skills in the course, which may have made the students aware of the importance of appreciative interaction with people as a prerequisite for a successful veterinary consultation. The competencies of “having the specialist knowledge at hand” and “being up to date with the latest knowledge” were predominantly given greater relevance by the students in the first years of study. The stronger focus of these students on specialist knowledge could result from the observation that students usually find it easier to acquire specialist knowledge than to deal with aspects of personal identity development [42,43]. This may have been counteracted by the practical communication course, as the third-year students after this course indicated the least relevance to these specialist knowledge-related competencies. The relevance of manual and technical skills was rated rather low overall but was rated highest by first-year students and decreased continuously until the rating of third-year students after the practical communication course, which was the lowest. Veterinarians’ ratings of the relevance of these competencies were lower than those of first-year students but higher than those of the other student groups surveyed. In another study, in which veterinary students and veterinarians were asked to indicate the importance of different skills for veterinary practice, technical and surgical skills were ranked higher by students than by veterinarians [44]. In the study presented here, this was only the case when comparing the results of first-year students and veterinarians. During the communication classes, manual and technical skills became decreasingly relevant. The competencies of “being tidy” and “using medical terminology” were rated as having very little relevance by all groups surveyed. The use of medical terminology was still attributed to the highest relevance by the students of the first year; in the following years, this competency was continuously attributed to a lower relevance. Since the importance of this competency was rated lowest by the veterinarians, the assessment showed a convergence with the veterinarians’ rating over the course of the study. This could be an effect of the communication courses, as it has been shown that teaching communication clarifies that explaining medical terminology in language that pet owners can understand is of much higher significance [45]. The competency of being good at mental arithmetic was attributed the least relevance for the professional success of veterinarians by all groups surveyed.

Overall, after completing the practical communication course, students placed a higher significance on the relevance of competencies in the area of communication than the other student groups and veterinarians surveyed. At the same time, it was also these students who predominantly assigned the lowest significance to non-communicative competencies compared to the other groups surveyed. Although a similar effect was not observed among students who had already taken the e-learning course on the basics of communication, it cannot be ruled out whether the prior awareness of the relevance of communication skills through the e-learning course contributed to the practical course having this effect. This is because all the students who had participated in the practical course had also completed the e-learning course beforehand. It is not surprising that the assessment of the relevance of communication skills was higher among students after the practical communication course than among veterinarians, as recent studies indicate insufficient communication skills among veterinarians and strongly recommend integrating the teaching of communication skills into the veterinary curriculum [4]. Since such a curriculum, and thus an obligatory teaching of communication skills, does not yet exist in Germany [18], it can be assumed that the majority of the veterinarians who participated in the survey had no communication training. However, based on their professional experience, veterinarians have an implicit understanding of communicative competencies [33]. This could explain why they still rated the relevance of communicative competencies higher than the students who had not yet completed the practical communication course.

Another aspect investigated was whether the experiences in the communication courses (e-learning and practical course) could influence students’ self-assessment of their communication skills. For this purpose, the SE-12 questionnaire, which was developed for measuring self-efficacy with regard to clinical communication skills in healthcare professionals [31], was adapted to veterinary medicine and used for the student survey. The analysis of the overall SE-12 scale showed no significant increase in self-evaluation of communication skills for the second-year students as well as the third-year students before the practical communication course. These students had taken the e-learning course on the basics of communication in the second semester of their first year of study. In two other studies, which also used the SE-12 questionnaire to assess the effects of online courses on communication, similar results were obtained [46,47]. The first of these studies examined the effect of an educational intervention on person-centered communication based predominantly on e-learning [48] for nursing assistants in home care [46]. They found no significant differences between outcomes measured before and after the intervention [46]. In the other study, a blended learning course for teaching patient communication to general practice residents was developed and evaluated at three different time points [47]. The first time of the survey was before the course, the second time after the e-learning section, and the third time after the practical training sessions [47]. This study also showed no significant increase in self-assessment of communication skills after the e-learning section [47]. However, the fact that the e-learning course did not have a significant effect on the self-assessment of communication skills in the study presented could also be because the students were not surveyed directly after the course, but more than three months (second-year students) or more than a year later (third-year students before the practical course). For instance, one study showed that among healthcare professionals, the significant increase in self-assessment of communication skills measured immediately after communication training decreased slightly but statistically significantly when the survey was repeated after six months [49]. This would also explain why the self-assessment of the third-year students before the practical communication course was lower on average than that of the second-year students. Third-year students after the practical communication course showed significantly higher self-assessments of communication skills than first-year students and third-year students before the practical course. This result is consistent with the findings of the study in which blended learning was used to teach communication skills; there too, a significant increase in self-assessment of communication skills was found only after practical training [47]. However, in a study in which communication skills were assessed following three different types of communication training (e-learning only, practical course only, and e-learning and practical course), it was found that while the combination of e-learning and practical course produced the best results, e-learning only had a higher effect than practical training only [50]. Therefore, also in the self-assessment of communication skills, it cannot be ruled out that the knowledge acquired in advance in the e-learning course could have had an influence on the significantly higher self-assessment of the students after the practical course.

The significantly higher self-assessment of communication skills among third-year students after the practical communication course in the total score was only confirmed for the results of six items of the SE-12 questionnaire. Third-year students’ self-assessment after the practical course regarding skills in structuring the consultation and checking owners’ understanding of the information was significantly higher than in all other student groups surveyed. Self-assessment in the skills of identifying the issues the owners wish to address, planning the conversation with the owners, and clarifying the knowledge of the owner were significantly higher among third-year students after the practical communication course than before the course, as well as among first-year students. For the skill of making a plan based on shared decisions, third-year students’ self-assessment was significantly higher after than before the practical course. All these skills were explicitly addressed in the practical communication course. Planning the conversation, structuring the consultation, and identifying the owner’s issues are elements of the CCOG [17] taught in the e-learning course and applied in the practical course during the anamnesis role plays. Clarifying the owner’s knowledge, checking the owner’s understanding, and making a plan based on shared decisions are elements of the SPIKES model [25] applied in the practical course during the equine asthma role play. No significant differences between student groups were found for the self-assessment in the skills of urging the owner to expand problems, listening without changing focus, encouraging the owner to express thoughts and feelings, showing appropriate nonverbal behavior, showing empathy, and closing the conversation. What is notable is that these skills are more about the relationship level. As studies with medical students have shown, little prior experience with consultations and thus few existing cognitive schemas lead to working memory resources being directed primarily toward cognitive processes and thus fewer resources being available for processing emotions [51]. Therefore, the lack of significant effects in the self-assessment of these skills after the practical communication course could be explained by the assumption that the working memory resources available to the students were predominantly bound by the cognitive processes required to implement the structural objectives in the role play, leaving them with few resources to perceive and attend to the emotional and relational aspects. With a stronger routine in the structural processes of conversation, students would have the space to focus more on the emotional aspects and subsequently feel more competent with these communication skills as well. For this purpose, it is necessary to expand the communication curriculum with further mainly practical courses.

Some limitations must be considered when interpreting the results. While the first-year, second-year, and third-year students were different cohorts of students, the same students were surveyed before and after the practical communication course. Therefore, the comparison of the cross-sectional and longitudinal data must be done carefully. On the one hand, the third-year students might have been focused on the relevance of communicative competencies by the survey before the practical communication course; on the other hand, knowing the questionnaire might have had an influence on the way they answered the questions after the practical course. This may have resulted in the attribution of the higher relevance of communicative competencies and the significantly higher self-assessment of communication skills. To test whether the repeated survey had an impact on the results, the survey should be delivered again to current first-year students at the same time in future years. It has been shown that self-assessment of communication skills yields favorable results immediately after communication training rather than several months later [49]. Therefore, the effects of the practical communication course might be overestimated since the survey was conducted immediately after the course, whereas the survey after the e-learning course on the basics of communication was conducted several months after completing the course. To take this fact into account, a follow-up survey of the students should be conducted six months after the practical course. In addition, when assessing the outcome of the e-learning course, it must be considered that although it was checked whether the pages had been worked on by the students, it was not possible to check how intense the attention and concentration and therefore the engagement with the content, was during this work. Moreover, the value of self-assessment as an instrument for measuring the outcome of communication courses must be critically viewed, because the assessment of communication skills by the students themselves may indeed turn out to be better than the assessment by the simulation person involved in the communication [52]. Consequently, it should be examined whether the effects of the practical communication course can also be replicated using third-party assessments or by examining communication skills, e.g., in an objectively structured clinical examination [53]. Finally, assessing veterinary communication skills using the SE-12 questionnaire [31] is limited to communication skills aimed at an interaction with pet owners that is comparable to the direct interaction between physician and patient. However, with the animal as the patient, the communication challenges for veterinarians go far beyond this and can range from being the pet’s advocate to being a service provider for the pet’s owner. Veterinarians do not only treat the animals and potentially deal with euthanasia, but also have to assess and discuss the financial situation and the compliance and skills of the owner to follow their treatment plans [37,38,54,55].

## 5. Conclusions

Teaching communication skills in the e-learning course and in the practical course resulted in different effects on students’ perceptions of competencies relevant to veterinary careers. The relevance of communicative competencies was rated significantly higher by the students after completion of the practical communication course than by the surveyed veterinarians as well as by the students who had not completed any communication course or only the e-learning course. Students who were surveyed after the practical communication course also scored significantly higher on the self-assessment of their communication skills than students who did not take communication courses or who only completed the e-learning course. However, since all students who participated in the practical communication course also completed the e-learning course beforehand, the significant effects on recognizing the relevance of communicative competencies and the positive influence on self-assessment of communication skills cannot be attributed exclusively to learning in the practical course. It remains to be investigated whether these outcomes of the communication courses can also be confirmed in a longitudinal study as well as through third-party assessments or in objectively structured examinations.

## Figures and Tables

**Figure 1 animals-13-01516-f001:**
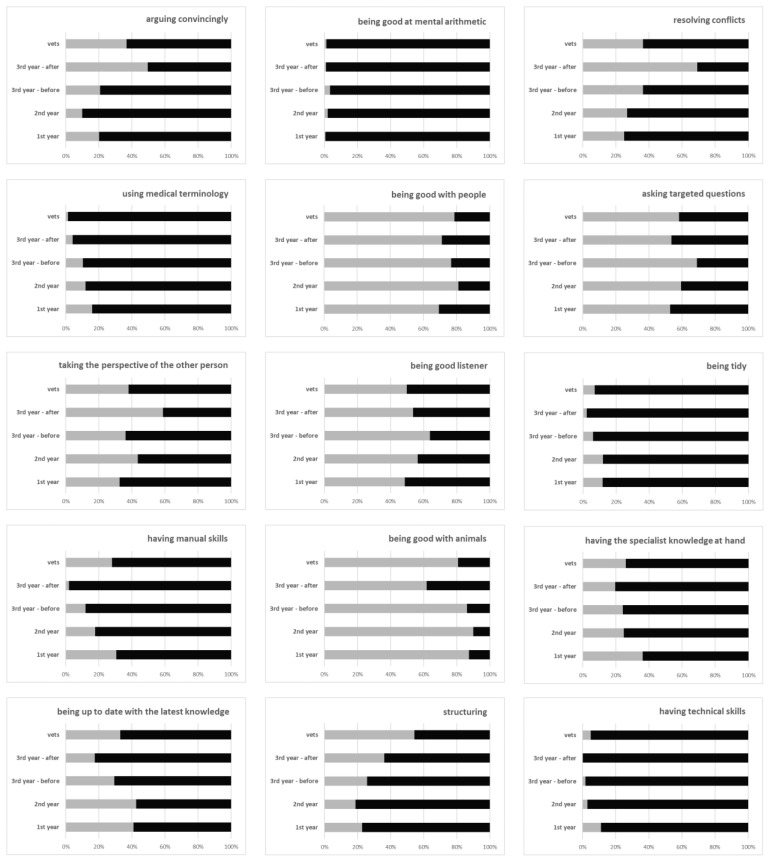
Assessment of the different competencies as relevant (grey = selected, black = not selected) for a successful veterinary career by first-year students (1st year, n = 144), second-year students (2nd year, n = 101), third-year students before (3rd year—before, n = 106) and after (3rd year—after, n = 97) the practical communication course, and veterinarians (vets, n = 160).

**Figure 2 animals-13-01516-f002:**
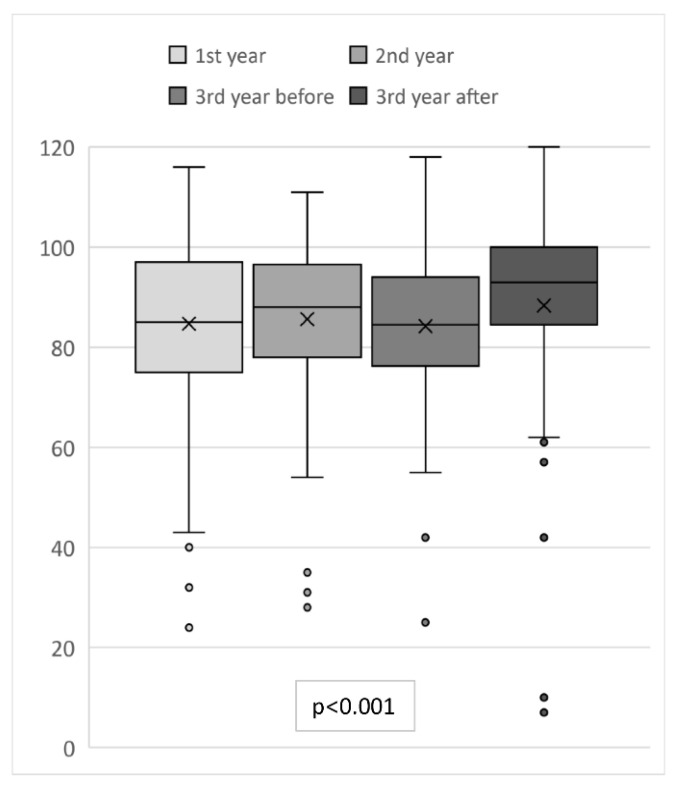
Self-assessment of communication skills by first-year students (1st year, n = 121), second-year students (2nd year, n = 93), and third-year students before (3rd year before, n = 108) and after (3rd year after, n = 97) the practical communication course using the SE-12 questionnaire [31] adapted to veterinary medicine, total score calculated from the sum of the results of the 12 items; cross: mean, whiskers: 10th and 90th percentile.

**Figure 3 animals-13-01516-f003:**
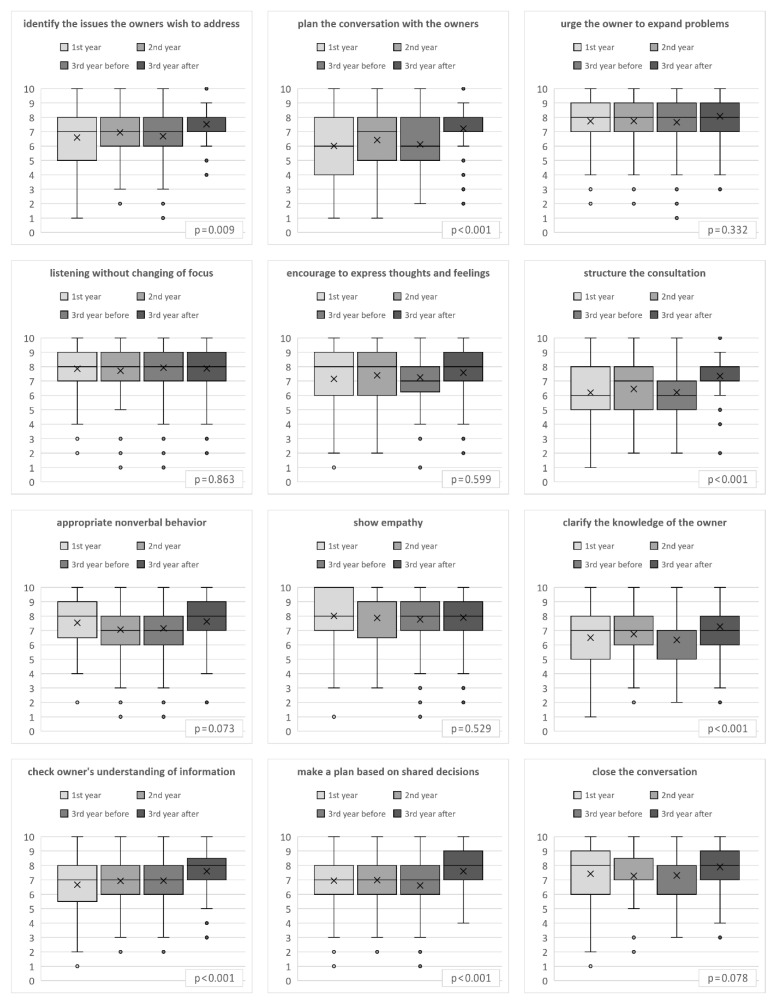
Self-assessment of communication skills by first-year students (1st year, n = 121), second-year students (2nd year, n = 93), and third-year students before (3rd year before, n = 108) and after (3rd year after, n = 97) the practical communication course using the SE-12 questionnaire [23] adapted to veterinary medicine, results for each of the communication skills defined in SE-12; cross: mean, whiskers: 10th and 90th percentile.

**Table 1 animals-13-01516-t001:** Questionnaire 1: Respondents were asked to select five competencies from the list of 15 that they consider most relevant to a successful career for veterinarians (translation of the original German statements).

For veterinarians to be successful in their careers, it is important that they…
… can argue convincingly.
… are good at mental arithmetic.
… can resolve conflicts.
… can use medical terminology correctly.
… are good with people.
… can ask targeted questions.
… can take the perspective of the other person.
… are good listeners.
… are tidy.
… have manual skills.
… are good with animals.
… always have the specialist knowledge at hand.
… are always up to date with the latest knowledge.
… can structure.
… have technical skills.

**Table 2 animals-13-01516-t002:** Questionnaire 2: SE-12 questionnaire for the self-assessment of self-efficacy regarding communication skills [31], adapted to veterinary medicine by replacing “patient” with “patient owners”. Respondents indicated their assessment on a 10-point Likert scale ranging from 1 (very uncertain) to 10 (very certain) or they indicated “not relevant”.

How certain are you that you are able to successfully…
… identify the issues the patient owners wish to address during the conversation?
… make an agenda/plan for the conversation with the patient owners?
… urge the patient owners to expand on his or her problems/worries?
… listen attentively without interrupting or changing of focus?
… encourage the patient owners to express thoughts and feelings?
… structure the conversation with the patient owners?
… demonstrate appropriate nonverbal behavior (eye contact, facial expression, placement, posture, and voicing)?
… show empathy and acknowledge the patient owners’ views and feelings?
… clarify what the patient owners know in order to communicate the right amount of information?
… check patient owners’ understanding of the information given?
… make a plan based on shared decisions between you and the patient owners?
… close the conversation by assuring, that the patient owners’ questions have been answered?

**Table 3 animals-13-01516-t003:** Differences concerning the relevance of competencies in the veterinary profession between different groups of veterinary students and veterinarians (chi-square test; n = 618; degrees of freedom = 4).

Competency	Chi-Square	*p*-Value
Arguing convincingly	52.9	<0.001
Resolving conflicts	54.4	<0.001
Using medical terminology	25.3	<0.001
Being good with people	6.6	0.158
Asking targeted questions	8.2	0.084
Taking the perspective of the other person	27.6	<0.001
Being good listener	25.7	<0.001
Being tidy	10.5	0.033
Having manual skills	41.1	<0.001
Being good with animals	36.2	<0.001
Having the specialist knowledge at hand	11.1	0.260
Being up to date with the latest knowledge	19.6	<0.001
Structuring	52.0	<0.001
Having technical skills	21.0	<0.001

**Table 4 animals-13-01516-t004:** Comparison of the self-assessment of communication skills between different groups of students (1st = first year students, n = 121; 2nd = second-year students, n = 93; 3rd(b) = third year students before the practical communication course, n = 108; 3rd(a) = third year students after the practical communication course, n = 97).

Communication Skill	Kruskal–Wallis Test		Post Hoc Tests (Adjusted *p*-Values)					
	Chi-Square	*p*-Value	1st to 2nd	1st to 3rd(b)	1st to 3rd(a)	2nd to 3rd(b)	2nd to 3rd(a)	3rd(b) to 3rd(a)
Identify the issues the patient owners wish to address	11.6	0.009	1.0	1.0	0.037	1.0	0.405	0.023
Make an agenda/plan for the conversation	22.3	<0.001	1.0	1.0	0.001	1.0	0.099	<0.001
Urge the patient owners to expand problems	3.4	0.332	Not performed					
Listen without changing of focus	0.7	0.863	Not performed					
Encourage to express thoughts and feelings	1.9	0.599	Not performed					
Structure the conversation	27.3	<0.001	1.0	1.0	<0.001	1.0	0.007	<0.001
Demonstrate appropriate nonverbal behavior	7.0	0.073	Not performed					
Show empathy	2.2	0.529	Not performed					
Clarify the patient owners’ knowledge	17.0	<0.001	1.0	1.0	0.008	1.0	0.226	0.001
Check patient owners’ understanding	16.3	<0.001	1.0	1.0	0.001	1.0	0.046	0.049
Make a plan based on shared decisions	16.6	<0.001	1.0	1.0	0.099	1.0	0.109	0.001
Close the conversation	6.8	0.078	Not performed					

## Data Availability

Data are contained within the article or Appendix A. The data presented in this study are available in the Appendix A.

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
