# Peer review of "Outcome of Communication Training in Veterinary Studies: Influence on the Perception of the Relevance of Veterinary Competencies and Self-Assessment of Communication Skills"

_animals, 2023, doi:10.3390/ani13091516_

Round 1

Reviewer 1 Report

“The survey of these two groups of students took place during the dissection courses in which all students of the semester participate.”

Unclear as to when this occurred during the timing of the curriculum -can the authors specify this?

The word “appeals” is used frequently in the manuscript, but it is unclear what the definition of “appeals” is – can the authors clarify? Outcome of communication trainings in veterinary studies: In-fluence on the perception of the relevance of veterinary competencies and self-assessment of communication skills

May be useful to better define LimeSurvey for individuals not familiar with this survey platform?

Author Response

Point 1: “The survey of these two groups of students took place during the dissection courses in which all students of the semester participate.”

Unclear as to when this occurred during the timing of the curriculum -can the authors specify this?

Response 1: Thank you very much for this advice. The information that the students were surveyed during the dissection courses in the first and third semesters, as well as the explanation that they were provided with the QR code for participation, has now been added.

Point 2: The word “appeals” is used frequently in the manuscript, but it is unclear what the definition of “appeals” is – can the authors clarify? Outcome of communication trainings in veterinary studies: In-fluence on the perception of the relevance of veterinary competencies and self-assessment of communication skills

Response 2: Thank you for pointing out that the reference to the appeal level of Schulz von Thun's 4-sides model is not clear. To clarify this, a sentence with the meaning of the appeal level has been added.

Point 3: May be useful to better define LimeSurvey for individuals not familiar with this survey platform?

Response 3: Thank you for the recommendation. It has now been added that LimeSurvey is an online survey application.

Reviewer 2 Report

This is a very interesting article on the highly topical issue of communication skills in veterinary medicine. It addresses the question of how veterinary students at different stages of their studies, as well as practicing veterinarians, perceive the relevance of different veterinary competencies and their individual communication skills. The basis of the examination are communication courses offered at Freie Universität Berlin in the first and third year of study. The goal was to determine if and how these courses influence perceptions of the relevance of veterinary competencies and self-assessments of communication skills.

The authors are to be commended on addressing an important and timely topic in the veterinary profession. Some suggestions to strengthen the readability and reliability of the findings can be found below.

Introduction

Since this article addresses such an important and timely topic, I would recommend emphasizing this more clearly in the introduction as well. From my point of view the current situation of veterinarians should be discussed here in more detail to underline the importance of the topic (e.g., general information on perceived stressors in veterinary practice, effects of these stressors, links between veterinary education and these stressors, etc.). Veterinarians are asked to do a lot in practice, and the gap between training and practice that still exists according to current state of research should be clarified.

Participants, Materials and Methods

·       As for the description of the courses, I would recommend focusing less on the specific approach of the modules and more on the skills learned, as this information is relevant to the methods section.

·       Some sentence beginnings in the document look like there are two spaces before the beginning.

·       The first paragraph in the description of the questionnaires is a bit irritating, because it is detailed information without knowing the used instruments.  

Results

Table 3: I would recommend to delete the competence "Good at mental arithmetic" from the table and only write the information regarding this item in the text.

Minor editing of English language required

Author Response

Point 1: Introduction: Since this article addresses such an important and timely topic, I would recommend emphasizing this more clearly in the introduction as well. From my point of view the current situation of veterinarians should be discussed here in more detail to underline the importance of the topic (e.g., general information on perceived stressors in veterinary practice, effects of these stressors, links between veterinary education and these stressors, etc.). Veterinarians are asked to do a lot in practice, and the gap between training and practice that still exists according to current state of research should be clarified.

Response 1: Many thanks for this valuable suggestion. The relationship between communication skills and stress experience, its consequences and the discrepancy between this knowledge and the lack of training in communication have now been addressed in the introduction.

Point 2: Participants, Materials and Methods: As for the description of the courses, I would recommend focusing less on the specific approach of the modules and more on the skills learned, as this information is relevant to the methods section.

Response 2: Thank you for this advice. In order to focus more on the skills that should be learned in the communication courses, the learning objectives have now been specified for the respective modules and lessons.

Point 3: Participants, Materials and Methods: Some sentence beginnings in the document look like there are two spaces before the beginning.

Response 3: The spaces are removed now.

Point 4: Participants, Materials and Methods: The first paragraph in the description of the questionnaires is a bit irritating, because it is detailed information without knowing the used instruments.

Response 4: We apologize for the confusion. The questions described in this paragraph for the veterinarians or students surveyed at the end of the practical course were at the beginning of the online survey before the questionnaires. This information has now been added.

Point 5: Results: Table 3: I would recommend to delete the competence "Good at mental arithmetic" from the table and only write the information regarding this item in the text.

Response 5: Thank you very much, we followed your suggestion.

Point 6: Minor editing of English language required

Response 6: Thank you very much for this advice. Our manuscript has now been finally proofread by a native speaker.

Reviewer 3 Report

The article presents the outcomes of a study on the effect of different teaching elements in veterinary education regarding communication skills.

The authors deal with a highly relevant issue in veterinary medicine and suggestions for implementations of curricular elements targeted at communication skills are, indeed, needed. Therefore, I consider the here presented research results as an important contribution to the field of veterinary education.

I only have a few questions and comments regarding this manuscript.

The self-assessment of communication competencies is based on a questionnaire from human medicine about physician-patient communication. Having some experience in research on similarities and differences in human and veterinary medicine, I wonder -  to what extent does the translation of this questionnaire cover the fact that relationships in veterinary medicine are triangular (patient - patient owner – vet) in contrast to the one-on-one relationship in human medicine (patient - physician)? After all, a communication-related challenge for veterinarians might evolve from the discrepancy between the role as the “animal’s/patient’s advocate” and the role as "service provider" to the client (i.e. the patient owner). Maybe, the authors can add an explanation regarding the transfer/translation?

Additionally, to me, the design and explanation of the figures on p. 8 are a bit confusing. At first glance, the black bars seem to represent the participants’ answers, but at second glance, the lighter bars indicate the selection of the skills by the participants. Furthermore, in the caption, it says “grey” means “yes”. If I understand the authors’ questionnaire correctly, there was no “yes” or “no” question but the task to identify the 5 most important skills among all 15 (which does not say that any of those that are not selected by a participant are “not” important, but just not as important as the five that are selected). As a result of such a questionnaire, I would rather expect a ranking of those skills that were judged to be most important to those that were judged to be least important. Maybe the authors can find a way to present the results in a less dichotomous (“yes/no”) way and more like a ranking? If that is not the basic idea, it would be good to explain why precisely 5 of 15 items had to be selected (instead of, e.g., ranking all 15 items according to their relevance). 

Furthermore, I am curious why the authors limited questionnaire 2 about the self-efficacy to student participants. To me, it would have been interesting to compare their answers to veterinarians’.

Finally, it seems remarkable to me that the “third year after intervention” group rates “being good with people” even more relevant than “being good with animals” which fits the authors’ point mentioned in the introduction that there might be a contrast between the initial motivation for becoming a vet and the actual outline of the job as a vet. Maybe that aspect could be added to the discussion.

Otherwise, I enjoyed reading the manuscript and I am looking forward to see it published.

Author Response

Point 1: The self-assessment of communication competencies is based on a questionnaire from human medicine about physician-patient communication. Having some experience in research on similarities and differences in human and veterinary medicine, I wonder -  to what extent does the translation of this questionnaire cover the fact that relationships in veterinary medicine are triangular (patient - patient owner – vet) in contrast to the one-on-one relationship in human medicine (patient - physician)? After all, a communication-related challenge for veterinarians might evolve from the discrepancy between the role as the “animal’s/patient’s advocate” and the role as "service provider" to the client (i.e. the patient owner). Maybe, the authors can add an explanation regarding the transfer/translation?

Response 1: Thank you very much for this feedback. You are of course right that communication with pet owners in the way of physician-patient communication represents only one aspect of veterinary communication. We decided to use the SE-12 questionnaire to have a valid tool to address the communication skills we have taught in our communication courses. However, we have now included into the limitations section that there are many more aspects to veterinary communication.

Point 2: Additionally, to me, the design and explanation of the figures on p. 8 are a bit confusing. At first glance, the black bars seem to represent the participants’ answers, but at second glance, the lighter bars indicate the selection of the skills by the participants. Furthermore, in the caption, it says “grey” means “yes”. If I understand the authors’ questionnaire correctly, there was no “yes” or “no” question but the task to identify the 5 most important skills among all 15 (which does not say that any of those that are not selected by a participant are “not” important, but just not as important as the five that are selected). As a result of such a questionnaire, I would rather expect a ranking of those skills that were judged to be most important to those that were judged to be least important. Maybe the authors can find a way to present the results in a less dichotomous (“yes/no”) way and more like a ranking? If that is not the basic idea, it would be good to explain why precisely 5 of 15 items had to be selected (instead of, e.g., ranking all 15 items according to their relevance).

Response 2: In fact, our original intention was to identify and discuss the top five competencies in the surveyed groups. But it turned out that the top five was approximately the same for all surveyed groups. However, when we regarded the results for the individual competencies, we found significant differences. Therefore, we decided to use this way of presentation. This explanation has now been added to the results section. And thank you very much for the remark about yes/no, that is indeed wrong. The graphs show the percentage of selected/not selected competencies. This has now been corrected in the figure legends.

Point 3: Furthermore, I am curious why the authors limited questionnaire 2 about the self-efficacy to student participants. To me, it would have been interesting to compare their answers to veterinarians’.

Response 3: Thank you very much for this question. We used the SE-12 questionnaire to test the effect of our communication courses on self-assessment of communication skills. Since the veterinarians surveyed did not participate in these communication courses, we did not use this questionnaire with the veterinarians. In addition, it can be assumed that the bias in self-assessment of communication skills caused by social desirability is particularly high among professional veterinarians, so that no meaningful results can be achieved. In assessing the relevance of the different veterinary competencies, we included the group of veterinarians because we wanted to compare the students' assessment with the veterinarians' assessment, assuming that the professional activities of veterinarians sensitize them to the relevance of communicative competencies. We have now added the latter aspect to the discussion to clarify why veterinarians were included in the survey of veterinary competencies. 

 Point 4: Finally, it seems remarkable to me that the “third year after intervention” group rates “being good with people” even more relevant than “being good with animals” which fits the authors’ point mentioned in the introduction that there might be a contrast between the initial motivation for becoming a vet and the actual outline of the job as a vet. Maybe that aspect could be added to the discussion.

Response 4: Thank you very much for this valuable advice. We have now added in the discussion that the third-year students after the practical course were the only group who ranked being good with animals lower than being good with people. Additionally, we added that this may be because the practical communication course may have made students aware of the importance of appreciative interaction with people for success in veterinary consultation.
